# MeshMoment: A Moment-Based Loss Function for 3D Learning on Meshes

## Abstract

In this paper, we address issues related to defining evaluation metrics and losses natively over meshes without resorting to intermediate representations. We demonstrate and discuss the pitfalls of commonly used intermediate representations - voxels and pointclouds - and propose a novel class of loss functions based on moments that can be calculated natively on meshes. The losses can be defined over both surfaces and volumes, and we investigate the vectorizability of computing moment losses and provide fully vectorized efficient implementations. We also expose their salient features which include differentiability, rigorous error bounds, and numerical stability. These loss functions can be applied towards various machine learning tasks like 3D reconstruction and Physics-Informed Neural Networks.

## 1 Introduction

Meshes stand out amongst a plethora of 3D object data structures as a flexible representation of surfaces and solids while striking a fine balance between precision and computational and storage costs. In the context of machine learning, various representations of objects have been utilized, not limited to meshes, point clouds, voxels, and parametric models.

Meshes have persisted as a preferred representation of 3D objects in computer graphics, and are practically necessary for scientific computing tasks e.g. simulation of physical systems. Although meshes are typically used to represent only the surfaces of objects in computer graphics, a mesh data structure can carry additional information e.g. scalar fields of pressure and vector fields of velocity in scientific computing over both surfaces and enclosed volumes. Therefore, it is desirable to define losses on meshes for scientific machine learning, of which a prime example is physics informed neural networks.

Despite its importance for representing 3D objects, there is a gap in metrics or loss functions defined natively over meshes for information represented using mesh data structures. Currently, there exist no losses that can operate natively on triangular meshes without converting to an approximate intermediate representation like point clouds or voxels (Wang et al., 2018; Gkioxari et al., 2019) .

3D shape reconstruction from images is a challenging task in computer vision. In recent years, deep learning-based approaches have produced remarkable results (Choy et al., 2016; Fan et al., 2017). These methods take as input single or multiple color images and predict 3D shapes as voxels (Choy et al., 2016), pointclouds (Fan et al., 2017), or triangular meshes (Wang et al., 2018). Pointclouds have been employed in a wide range of applications (Achlioptas et al., 2018; Charles et al., 2017; Qi et al., 2017). However, the unordered and irregular data format makes it challenging to define a similarity metric between two point clouds (Tong et al., 2021). Although defining similarity metrics on voxel grids can be straightforward (e.g. $L_2$ loss), the reconstruction quality of voxels is subpar unless computationally expensive and memory intensive fine grids are used. Triangle meshes on the other hand, offer a much higher-quality 3D shape representation than point clouds and voxel grids (Wang et al., 2018).

Moreover, pointclouds generate using sampling operations lose important surface details and are time-consuming (Wang et al., 2018). Losses computed over pointclouds sampled from identical meshes do not yield a loss of zero, even when the loss is averaged over multiple pointclouds. This

is especially evident in pathological cases, such as thin sheets or objects with intricate and complex geometry.

In this paper, we address this critical gap by introducing MeshMoment, a novel class of loss functions defined directly over triangular meshes using geometric moments. Unlike existing methods that depend on sampled point clouds or voxel-based representations, our moment-based loss operates natively on mesh surfaces, preserving geometric details and ensuring consistency across mesh triangulations. We focus on the case where the information represented by the mesh data structure is the surface of a 3D object and the computational task is 3D object reconstruction (e.g. from 2D images). The ideas developed in this paper are readily extendable to arbitrary numbers of dimensions and other application domains.

Geometric moments (GMs) and their orthogonal representations, e.g. Zernike moments (ZMs), Legendre polynomials, and Chebyshev polynomials, have been used in image processing and classification problems for decades (Hu, 1962; Teague, 1980). In particular, ZMs form an orthonormal basis, allowing easy reconstruction of the image. While GMs can be made invariant to translation and scaling, ZMs can also be made invariant to rotation (Khotanzad & Lu, 1989) which makes them ideal features for shape descriptors in shape classification problems (Vorobyov, 2011). Zernike moments have been successfully applied to 3D objects. The 3D-AF-Surfer model (Aderinwale, 2022) takes 3D ZMs as inputs to a deep neural network to retrieve proteins of the same fold more accurately than direct comparison of ZMs. A moment-based loss function was proposed by Jhanwar et al. (2022) over voxels for 3D dose distribution of radiation for radiation therapy. However, to our knowledge, 3D moments have not been used to describe a loss function over meshes or for 3D reconstruction.

The orthonormal property of Zernike moments allows computation of the approximation error using Parseval's identity. Although Zernike moments in general form an infinite sequence for exact representation, the error of a truncated sequence converges rapidly as terms are added.

However, computational cost, precision, and stability must be considered to apply GMs and ZMs towards training deep neural networks for 3D shape generation (Pozo et al., 2011; Houdayer & Koehl, 2022a). The calculation of higher order Zernike moments suffers from numerical instabilities. Recent work has demonstrated stable procedures, but relies on expensive quadrature methods. A naive implementation of a moment-based loss requires prohibitively high compute time and memory. Standard backpropagation algorithms are incapable of handling backpropagation through moment calculations since the computational graph exceeds memory limits even for simple meshes. In this work, we introduce a fully differentiable moment-based loss and its efficient implementation on GPUs. We provide efficient procedures to calculate moments on triangle meshes, along with an explicit formulation and procedure to compute gradients. Our optimized implementation completely resolves memory issues and reduces compute time by several orders of magnitude.

Since 3D moments can be calculated directly from meshes (Houdayer & Koehl, 2022b), voxels (Deng & Gwo, 2020), and pointclouds (Novotni & Klein, 2004), they are convenient to use and enable usage of datasets with a mixture of 3D representations without introducing errors arising from conversion to an intermediate representation. These properties of GMs and ZMs make them attractive for use as **both a loss** and **an evaluation metric** for tasks in machine learning with mesh data structures.

## 2 PRELIMINARIES

Chamfer Distance (CD) is a popular and widely used metric to compare 3D objects. A 3D mesh can be represented as a point cloud by densely sampling points over the surface of the mesh. Given two point clouds $P$, $Q$ with normal vectors. Let $\Lambda_{P,Q} = \{(p, argmin_q||p - q||) : p \in P\}$ be the set $(p, q)$ such that $q$ is the nearest neighbour of $p$ in $Q$, with normal vectors $u_p, q_q$. The chamfer distance between point clouds $P$ and $Q$ is

$$\mathcal{L}_{cham}(P, Q) = \frac{1}{|P|} \sum_{(p,q)\in\Lambda_{P,Q}} ||p - q||^2 + \frac{1}{|Q|} \sum_{(q,p)\in\Lambda_{Q,P}} ||q - p||^2 \tag{1}$$

Similarly, an absolute normal distance is

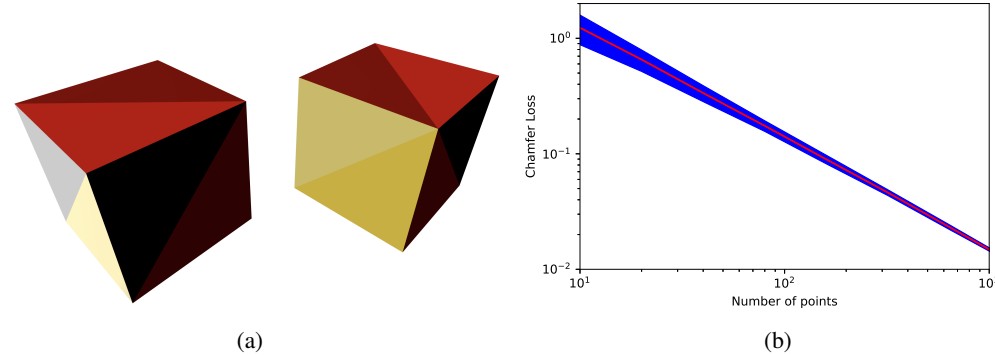

(a)             (b)

Figure 1: (a) Identical cubes with different mesh triangulations. Moments are invariant to equivalent representations. (b) Convergence of chamfer loss for various numbers of points, 100 pairs of pointcloud each, sampled over the same cube bounded by $x, y, z \in [-1, 1]$ shown with a confidence interval of 1 standard deviation.

$$\mathcal{L}_{norm}(P, Q) = -\frac{1}{|P|} \sum_{(p,q) \in \Lambda_{P,Q}} |u_p \cdot u_q| - \frac{1}{|Q|} \sum_{(q,p) \in \Lambda_{Q,P}} |u_q \cdot u_p| \qquad (2)$$

The chamfer and normal losses penalize a mismatch in the distribution of positions and normals between two pointclouds. Naively minimizing these losses leads to degenerate meshes (Gkioxari et al., 2019). A remedy is to either increase the sample size or add shape regularizers. Edge loss has demonstrated acceptable performance in remedying mesh degeneracy issues, and is defined as:

$$\mathcal{L}_{edge}(V, E) = \frac{1}{|E|} \sum (v, v') \in E||v - v'||^2 \qquad (3)$$

where $E \subseteq V \times V$ are the edges of the predicted mesh. Edge loss penalizes long edges, which are commonly seen in degenerate meshes when chamfer and normal distances are minimized. Another common loss term is Laplacian loss, which encourages smooth surfaces.

Although chamfer and normal losses over point clouds are straightforward to implement, their performance is limited by the number of points sampled. Pointclouds have trouble capturing fine details like slender features or sharp corners in objects, and chamfer losses over pointclouds sampled over the same object are neither expected to be zero, nor are they consistent. In fact, a large sample of chamfer losses of point clouds of the same size over the same object follows a distribution with a non-zero mean. To improve the performance of pointclouds, exponentially many points need to be sampled. This implies that the bottleneck for reconstruction quality is in fact the effectiveness of point clouds in differentiating between two identical objects . We provide evidence of these issues in Fig. 1b. Since chamfer and normal losses require finding nearest neighbours between two point clouds, they require $O(N^2)$ operations, further exacerbating the requisite computational cost for improved performance. For voxel representations, the loss is typically defined using MSE. Although the MSE loss over voxels does quantify the degree of mismatch between two voxel representations, the location of the mismatch is not taken into account. Alternatively, orthogonal 3D moments of an object are more sensitive to such perturbations and exhibit a more favorable asymptotic scaling for object reconstruction, scaling polynomially with the number of moments computed and linearly with the number of mesh faces.

Current approaches use a combination of different losses such as vertex loss, 3D joint loss, 2D joint loss, and heatmap loss (Kim et al., 2023). A loss based on moments could be used either as a standalone loss or an additional loss to capture fine details and introduce invariance to the overall loss.

## 3 MOMENTS AND MOMENT LOSS

**Moments from 3D shapes**. Consider a scalar field $f(\mathbf{p})$ defined over $\mathbf{p} \in \Omega \subset \mathbb{R}^3$ and a collection of functions $M_I(\mathbf{p})$. Moments of this scalar field with respect to the functions are defined as

$$\mathcal{M}_I(f) \equiv \int_\Omega f(\mathbf{p})M_I(\mathbf{p})d\Omega \qquad (4)$$

For 3D shapes $f(\mathbf{p})$ may be defined as the indicator function ($f(\mathbf{p}) = 1$ over either the volume or surface of the object) to obtain volume-like or surface-like moments denoted as $f^{(V)}(\mathbf{p})$ and $f^{(S)}(\mathbf{p})$ respectively. The computation of surface and volume moments follows a similar procedure. Geometric and orthogonal moments over volumes or surfaces are classic tools for image and 3D object recognition, characterization, and reconstruction. The calculation of volume-like moments requires the mesh representation to follow a consistent winding order and the mesh must also be watertight, which is rarely the case for realistic datasets and requires expensive mesh repair procedures for non-watertight meshes. Since surface-like moments are not subject to such constraints, they are considered exclusively in this paper.

The simplest moments are geometric moments (GM), for which

$$M_I(\mathbf{p}) \equiv G_{ijk}^{(S)}(\mathbf{p}) = x^i y^j x^k \qquad (5)$$

using which we may form a rank 3 tensor $\mathcal{G}_{i,j,k}^{(S)}$ of surface GMs. The order of a moment is $m = i + j + k$.

Moments can be made invariant to translation using central moments defined using central coordinates

$$\mathbf{p} - \bar{\mathbf{p}} = (x - \bar{x}, y - \bar{y}, z - \bar{z}) \qquad (6)$$

where $\bar{x}, \bar{y}, \bar{z}$ are $\frac{\mathcal{G}_{1,0,0}^{(\cdot)}(\mathbf{p})}{\mathcal{G}_{0,0,0}^{(\cdot)}(\mathbf{p})}, \frac{\mathcal{G}_{0,1,0}^{(\cdot)}(\mathbf{p})}{\mathcal{G}_{0,0,0}^{(\cdot)}(\mathbf{p})}, \frac{\mathcal{G}_{0,0,1}^{(\cdot)}(\mathbf{p})}{\mathcal{G}_{0,0,0}^{(\cdot)}(\mathbf{p})}$. Moments can also be made invariant to scale by scaling the coordinates of the vertices as $\frac{V}{\mathcal{G}_{0,0,0}^{(\cdot)}(\mathbf{p})}$.

The sequence of GMs captures the overall shape of an object, and will be equal for identical objects regardless of mesh triangulation. However, GMs do not form an orthogonal representation and therefore their application towards reconstruction is not straightforward. As a remedy GMs can be transformed into orthogonal moments.

The most commonly used and successfully applied orthogonal moments are Zernike or Zernike-Canterakis moments (ZM) in 2D or 3D respectively. ZMs in 3D are defined using

$$M_I(\mathbf{p}) \equiv \bar{Z}_{nl}^m(\mathbf{p}) = R_{nl}(r)Y_l^m(\theta, \phi) \qquad (7)$$

where the overline denotes complex conjugation. $(r, \theta, \phi)$ are spherical coordinates in $\Omega$. Using this definition, we may form a rank 3 tensor $\mathcal{Z}_{l,m,n}^{(S))}$ of surface Zernike moments. In addition to scale and translation invariance, ZMs can also be made rotation invariant. ZMs demonstrate superior performance compared to other orthogonal moments. GMs can be transformed into ZMs using a linear transformation. However this transformation becomes numerically unstable for computing moments beyond $m = 15$ (Houdayer & Koehl, 2022b). ZMs may be calculated natively on a mesh without computing GMs but either require prohibitively expensive numerical quadrature computations (Houdayer & Koehl, 2022b) or approximations. Practical computation of derivatives of moments requires analytical gradients, which are complex for Zernike moments. Due to these difficulties and the fact that efficient and stable computation of Zernike moments is an active area of research, we do not consider Zernike moments and defer development of efficient and stable procedures for calculation of mesh losses using Zernike moments to future work.

We now define MeshMoment loss as

$$\mathcal{L}_\mathcal{M} = \frac{1}{2}\|\mathcal{M} - \tilde{\mathcal{M}}\|_2^2 \qquad (8)$$

where $\mathcal{M}$ and $\tilde{\mathcal{M}}$ represent the tensor moments for prediction and ground truth, respectively. These tensors may be truncated up to a desired order $m$.

Table 1: Comparison of 3D loss functions across desirable properties. Our proposed moment loss, based on surface geometric moments, is the only one to satisfy all five criteria. *indirectly responds to uneven sampling density, **local density awareness.

| Metrics | Task | Consistent | Efficient | Density-aware | Bounded | Smooth |
|---------|------|------------|-----------|---------------|---------|--------|
| Chamfer | Nearest neighbour | | | | | ✓ |
| Normal | Nearest neighbour | | | | | ✓ |
| Edge | Shape regularizer | ✓ | ✓ | * | | ✓ |
| Laplacian | Shape regularizer | ✓ | ✓ | ** | | ✓ |
| (Ours) | Geometry-aware | ✓ | ✓ | ✓ | ✓ | ✓ |

Although moments form infinite sequences, they rapidly characterize the overall shape of an object. In fact, Parseval's identity for orthonormal functions provides a bound on the error when using orthogonal moments:

$$\|f(\mathbf{p})\|^2 = \int_\Omega f(\mathbf{p})^2 d\Omega = \sum_{l,m,n} \|\mathcal{M}^Z_{l,m,n}\|^2 \tag{9}$$

$$\epsilon = \|f(\mathbf{p})\|^2 - \sum_{l,m,n} \|\mathcal{M}^Z_{l,m,n}\|^2 \tag{10}$$

where $\int_\Omega f(\mathbf{p})^2 d\Omega$ is simply the surface area integral for surface moments, which is easily computable for meshes. Zernike moments are well-known to converge rapidly for smooth objects, i.e. the object does not have sharp corners or edges.

Since moments over meshes are analytic functions, they are differentiable, allowing training without the ELBO loss required for point-cloud samples.

**Loss Properties**. We summarize and compare the properties of commonly used 3D loss function in Table 1. While Chamfer and normal losses are widely used in mesh and pointcloud-based reconstruction, they exhibit significant drawbacks. Both rely on nearest-neighbor search, which introduces inconsistency, loss values remain non-zero even when comparing pointclouds sampled from the same mesh (See Figure 1a). These losses are also sensitive to sampling density and require $O(N^2)$ operations, limiting scalability. Moreover, they are unbounded, prone to local minima and often lead to degenerate reconstructions unless supplemented with regularizers. As a result, edge and Laplacian losses are typically introduced as regularizers to address these degeneracies. They encourage mesh smoothness and regularity but do not measure geometric similarity directly. Although they are efficient and differentiable, they do not capture global shape information and are not inherently bounded or density-aware.

In contrast, our proposed moment-based loss operates directly on the geometry of the object rather than sampled point representations. It is consistent across mesh triangulations, efficient under vectorized GPU computation, and inherently density-aware due to global shape integration. Unlike Chamfer or normal loss, it is bounded through Parseval's identity and smooth due to its analytic formulation. These properties make moment loss a strong candidate for both training and evaluation in mesh-based 3D reconstruction tasks.

## 4 COMPUTATION OF GEOMETRIC MOMENTS AND DERIVATIVES

There exist efficient and stable algorithms for calculation of surface-like and volume-like GMs naively on a mesh. A fully vectorizable, albeit suboptimal, $O(Fm^6)$ algorithm was proposed by Pozo et al. (2011) which was subsequently improved to a recursive $O(Fm^3)$ FLOPS algorithm by Koehl (2012), where $F$ is the number of faces and $m = i + j + k$ is the maximum order to which the moments are truncated. In our study, we first investigate these two techniques to investigate their scaling behavior for an optimized and vectorized implementation on GPUs. Using scaling data, we identify the candidate algorithm for scalability. Our numerical experiments are conducted on Nvidia RTX 4060, V100, and GH200 GPUs.

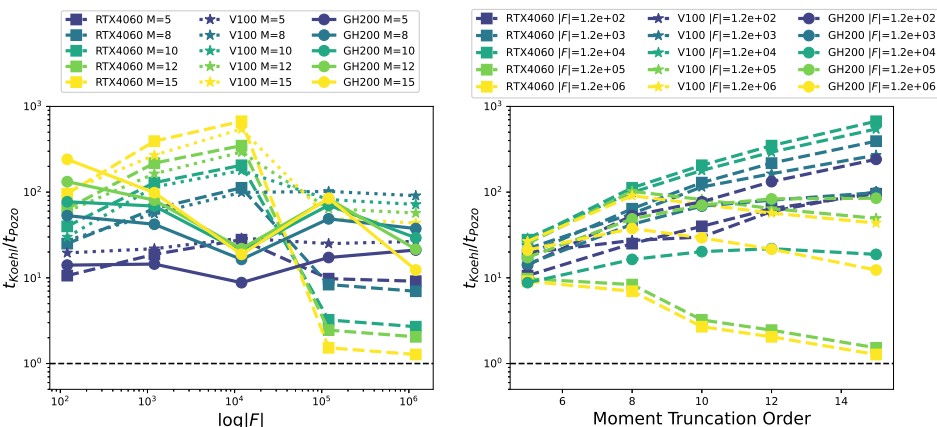

Figure 2: Comparison of optimized GPU implementations of Pozo and Koehl's algorithms for Top: increasing numbers of mesh faces Bottom: increasing moment truncation orders. Pozo's fully vectorized method exhibits significant speedup as the number of moments grow, and is throttled by increasing the number of mesh faces.

We implement optimized versions of Pozo and Koehl's algorithms to investigate scaling performance as shown in Figure 2. Our results indicate that even though Pozo's method is suboptimal in FLOP count, it runs up to 100 times faster on the GPUs used. The performance of Pozo's algorithm may be attributed to our highly efficient fully vectorized "einsum" implementation of operations in Pytorch, compared to Koehl's algorithm requiring at least three nested loops, which are not conducive for GPU performance. We note that as the number of faces increases, GPU performance is throttled. On the other hand, as the number of moments grows, the speedup improves for Pozo's algorithm. Since the computations of moments are almost embarrassingly parallel over the faces, the $\sim 100 - 1000\times$ speedup can be maintained by distributing the computations over multiple GPUs. In our experiments using RTX 4060 and V100 GPUs, we process 1000 faces in parallel, looping until all faces are processed. For the GH200 GPU, due to higher memory availability, we process 10,000 faces in parallel for each loop. For all experiments, we compute GM tensors truncated up to $m = 15$. Due to the non-orthogonality of GMs, higher order computations typically lead to imprecise results due to floating-point truncation errors. However, this issue is not expected to persist for the computation of orthogonal moments.

Based on numerical evidence, we conclude that Pozo's algorithm is better suited for vectorized computation of moments. The vectorized version of Pozo's algorithm is provided as Algorithm 1.

There exist various methods to approximate GMs and ZMs for improved computational efficiency, e.g. quadrature methods for approximate integration (Houdayer & Koehl, 2022b). However, we defer a numerical analysis of those methods to further studies. We also note that moments for the ground truth can be computed and stored efficiently in a pre-processing step, unlike point clouds which are typically sampled for both the ground truth and prediction on-the-fly.

We now turn our attention to issues related to gradient computation. The computational graph for moment calculation is highly complex and a naive usage of automatic differentiation engines very quickly leads to extremely high memory requirements. We note that beyond an observed speedup, Pozo's algorithm also admits a relatively simpler chain rule for backpropagation and does not require storing any intermediate computations as context for backpropagation. We therefore implement an optimized version of Pozo's algorithm with a manual implementation of the adjoint operator for derivatives, according to Algorithm 2.

We now formally state the computational procedure for computing the surface GMs of a mesh. Subsequently, we derive the chain rule for computing gradients of a GM loss. GMs are fully differentiable w.r.t. to the vertices of the mesh. We provide here also a full description of an efficient calculation of the derivatives of moments.

Given the faces and vertices of a triangle mesh as $(F, V)$ s.t. $F : \{\phi_a\}$ where $\phi : (x, y, z)$, $a \in \{1, 2, 3\}$, and $\phi \in V$, the calculation of surface GMs (Pozo et al., 2011) is defined as

$$\mathcal{G}_{ijk}^{(S)} = \sum_F 2 Area_F \mathcal{S}_{ijk} \tag{11}$$

where $Area_F = \frac{1}{2}\|\phi_{13} \wedge \phi_{23}\|_2$, $\phi_{ij} = \phi_i - \phi_j$

$$\mathcal{S}_{ijk} = \frac{i!j!k!}{(i + j + k + 2)!} \sum_{\substack{i_1+i_2+i_3=i \\ 0 \le i_1, i_2, i_3}} \sum_{\substack{j_1+j_2+j_3=j \\ 0 \le j_1, j_2, j_3}} \sum_{\substack{k_1+k_2+k_3=k \\ 0 \le k_1, k_2, k_3}}$$
$$\times (i_1|i_2|i_3)(j_1|j_2|j_3)(k_1|k_2|k_3)$$
$$\times x_1^{i_1} y_1^{j_1} z_1^{k_1} x_2^{i_2} y_2^{j_2} z_2^{k_2} x_3^{i_3} y_3^{j_3} z_3^{k_3} \tag{12}$$

where

$$(a|b|c) = \frac{(a + b + c)!}{a!b!c!} \tag{13}$$

$\mathcal{S}_{ijk}$ are 3D convolutions $(*)$ in terms of intermediate tensors

$$C_{ijk}^{(a)} = (i_a|j_a|k_a) x_a^{i_a} y_a^{j_a} z_a^{k_a} \tag{14}$$

so that

$$\mathcal{S}_{ijk} = \frac{i!j!k!}{(n + 2)!} C_{ijk}^{(1)} * C_{ijk}^{(2)} * C_{ijk}^{(3)} \tag{15}$$

We define a GM loss as

$$\mathcal{L}_\mathcal{G} = \frac{1}{2}\|\mathcal{G} - \tilde{\mathcal{G}}\|_2^2 \tag{16}$$

Using the chain rule we define the derivatives with respect to the vertices $\{\phi_a\}$ of each triangular face in $\{F\}$.

$$\frac{\partial \mathcal{L}_\mathcal{G}}{\partial \phi_a} = \frac{\partial \mathcal{L}_\mathcal{G}}{\partial \mathcal{G}} \frac{\partial \mathcal{G}}{\partial \phi_a} \quad \text{and} \quad \frac{\partial \mathcal{L}_\mathcal{G}}{\partial \mathcal{G}} = \mathcal{G} - \tilde{\mathcal{G}} \tag{17}$$

$$\frac{\partial \mathcal{G}_{ijk}}{\partial \phi_a} = \sum_{\{F\}} \frac{\partial\|\phi_{13} \wedge \phi_{23}\|_2}{\partial \phi_a} \mathcal{S}_{ijk} + \sum_{\{F\}} \|\phi_{13} \wedge \phi_{23}\|_2 \frac{\partial \mathcal{S}_{ijk}}{\partial \phi_a} \quad \text{where} \tag{18}$$

$$\frac{\partial \mathcal{S}_{ijk}}{\partial \phi_a} = \frac{\partial C_{ijk}^{(a)}}{\partial \phi_a} * C_{ijk}^{(b)} * C_{ijk}^{(c)} \quad \text{s.t.} \quad a \ne b \ne c \quad \text{and} \tag{19}$$

$$\frac{\partial C_{ijk}^{(a)}}{\partial \phi_a} = \begin{cases} 0 & \forall \phi_a' = 0 \\ \frac{\phi_a'}{\phi_a} C_{ijk}^{(a)} & \text{otherwise} \end{cases} \tag{20}$$

where $x_a', y_a', z_a' = i, j, k$, and

$$\frac{\partial\|\phi_{13} \wedge \phi_{23}\|_2}{\partial \phi_a} = \frac{(\phi_{13} \wedge \phi_{23})^T \frac{\partial(\phi_{13} \wedge \phi_{23})}{\partial \phi_a}}{\|\phi_{13} \wedge \phi_{23}\|_2}. \quad \text{where} \tag{21}$$

$$\frac{\partial(\phi_{13} \wedge \phi_{23})}{\partial \phi_a} = \tilde{\phi}_a^{(1)} \wedge \phi_{23} + \phi_{13} \wedge \tilde{\phi}_a^{(2)} \quad \text{according to} \tag{22}$$

| $a \setminus \phi$ | $\tilde{\phi}_a^{(1)}$ | | | $\tilde{\phi}_a^{(2)}$ | | |
|---|---|---|---|---|---|---|
| | x | y | z | x | y | z |
| 1 | $e_1$ | $e_2$ | $e_3$ | 0 | 0 | 0 |
| 2 | 0 | 0 | 0 | $e_1$ | $e_2$ | $e_3$ |
| 3 | $-e_1$ | $-e_2$ | $-e_3$ | $-e_1$ | $-e_2$ | $-e_3$ |

where $e_\phi \in \mathbb{R}^3$ are the standard basis vectors.

The 3D convolution operations can be a significant computational bottleneck unless computed carefully. In our implementation, the usage of the "einsum" contraction method for convolutions is critical for fast GPU implementation. For this we define the convolution contraction kernel defined as:

$$K_{ijk} = \begin{cases} 1 & \forall \, |j-k| == i \\ 0 & \text{otherwise} \end{cases} \tag{23}$$

We utilize the *opteinsum* library to optimize the contraction sequence.

Once the derivatives w.r.t. the vertices of each face $\frac{\partial \mathcal{L}_\mathcal{G}}{\partial \phi_a}$ are known individually, they can be collected using the face matrix $F$ to obtain the partial derivatives $\frac{\partial \mathcal{L}_\mathcal{G}}{\partial V}$ w.r.t. the vertices of the mesh. One again, we utilize the *opteinsum* library to optimize the contraction sequence for collecting the derivatives.

The procedure for computing the derivatives is provided in 2. We compare the time required for forward and backward passes for various mesh sizes and moment orders. As illustrated in Fig. 3, the backward pass time converges to $\approx 8\times$ forward pass time, exhibiting $O(1)$ overhead for gradient computations.

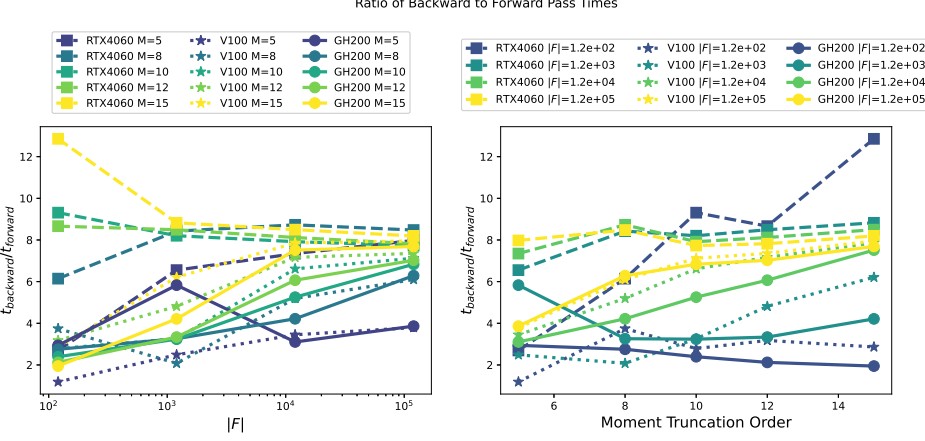

Figure 3: Ratio of forward times to backward times for our vectorized implementations for Top: increasing numbers of mesh faces Bottom: increasing moment truncation orders.

We now investigate the scalability of computing GMs compared to the number of faces. Forward and backward computation times are shown in 4. As expected, the computation times for both forward and backward passes scale linearly with the number of moments. For our implementation, the scaling with the moment order is quadratic. As such, we see that computation times scale polynomial in both the number of faces and order of moments.

Implementations of our vectorized algorithms for Koehl's forward pass and Pozo's forward and backwards passes are provided as supplementary material.

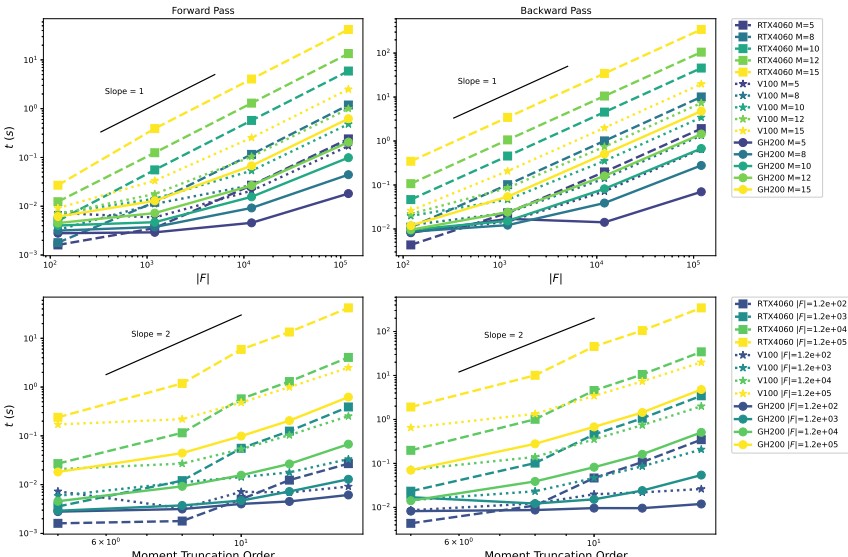

Figure 4: Scaling of forward and backward computations of moments for Top: increasing numbers of mesh faces Bottom: increasing moment truncation orders Left: forward pass Right: backward pass.

## 5 FUTURE WORK

Numerically stable utilization of moments as loss functions requires several considerations. GMs may be converted to orthogonal representations, e.g. Zernike and Legendre polynomial representations, using a linear transformation (which is trivially differentiable). However, this transformation is typically unstable (Houdayer & Koehl, 2022b) and it has been proposed to directly compute orthogonal representations instead of using GMs as an intermediate representation. This requires development of new efficient algorithms for computing orthogonal moments and their adjoints.

Recent work has also posited the approximate computation of moments (Houdayer & Koehl, 2022b) using quadrature rules. Although this would lead to a loss of mathematical properties, e.g. Parseval's identity, they may still be applicable as loss functions.

Beyond efficient computations of moments, whether moment functions can be used as a standalone loss or in conjunction with existing loss functions remains to be investigated.

## CONCLUSION

In this paper we have investigated the efficient computation of surface GMs for meshes and have proposed using moments as a loss function for data defined over meshes. While GMs are not orthogonal, and therefore not suitable for optimization e.g. as a loss function, they may serve as a metric for evaluating the performance of machine learning tasks related to meshes. Further work is required to develop efficient implementations of orthogonal moment representations e.g. Zernike moments to make them conducive for optimization tasks and leverage their properties of size and rotation invariance as needed.

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

## A    RELATED WORK

**3D shape representation**: There is a rich literature on 3D shape representation (Chen & Zhang, 2019; Mescheder et al., 2019; Niemeyer et al., 2020; Park et al., 2019; Sommer et al., 2022; Wu et al., 2015; Yariv et al., 2020) which falls into one of the two families: explicit representations such as pointcloud (Achlioptas et al., 2018; Charles et al., 2017; Qi et al., 2017), voxels (Girdhar et al., 2016; Wu et al., 2016) or meshes (Wang et al., 2018), and implicit functions such as signed distance function (SDF) (Park et al., 2019; Sommer et al., 2022).

Voxels are the 3D counterpart of pixel representation in 2D space. Voxel-based network architectures are easily constructed by extending image-based operations and are widely adopted by various methods (Girdhar et al., 2016; Wu et al., 2016; 2015). However, voxels suffer from high memory requirements and computational complexity, limiting their scalability for large-scale applications. Point cloud representations, which offer more compact and flexible representations, are well suited for large-scale neural networks (Achlioptas et al., 2018; Charles et al., 2017; Qi et al., 2017). However, they lack spatial order and arrangement which limits their use in traditional deep learning. Nonetheless, the issue can be addressed using PointCNN (Li et al., 2018) or GraphCNN (Wang et al., 2018). Like pointclouds, 3D triangular meshes are an accurate discrete 3D shape representation (Wang et al., 2018; Yuan et al., 2020; Gkioxari et al., 2019). They provide information about discrete patches and coordinates for each vertex and constitute a piece-wise linear approximation of the entire shape surface, serving as a higher-quality 3D shape than pointclouds. Moreover, meshes require less memory than voxels on the same resolution as they only represent the boundary surface rather than the entire volume.

Implicit representations use implicit functions (e.g., occupancy function (ON) (Mescheder et al., 2019) or SDF (Park et al., 2019; Sommer et al., 2022)) to describe 3D shape (Chen & Zhang, 2019;

Shim et al., 2023; Niemeyer et al., 2020; Oechsle et al., 2021; Yariv et al., 2020). SDF, in particular, computes surface normals and encodes, in a 3D space, each point distance to the nearest surface. However, they are slow as they require several network operations and post-processing to convert shapes into their explicit representation. In this paper, we adopt mesh representations to model 3D shape details without much effort.

**3D shape generation**. There are numerous frameworks for 3D generation (Achlioptas et al., 2018; Chen et al., 2021; Chen & Zhang, 2019; Gao et al., 2021; Ibing et al., 2023; 2021; Klokov et al., 2020; Liao et al., 2020; Nash et al., 2020; Pavllo et al., 2021; Sanghi et al., 2022; Shu et al., 2019; Tan et al., 2018) which can be divided into two main categories: encoder-decoder (Cheng et al., 2023) and GAN-based models (Zheng et al., 2022; Pavllo et al., 2021; Chen et al., 2021; Goodfellow et al., 2014). Some models, however, share elements of both categories.

GAN-based 3D methods such as in Zheng et al. (2022) extend StyleGAN for 3D shape by proposing global and local discriminator that enables GAN discriminator on SDF. In addition, Mittal et al. (2022) and Yan et al. (2022) use voxel for conditional 3D shape generation through a two-stage. Their auto-encoder based approach convert voxels into compressed latent space and generate 3D shapes using autoregressive model. Similarly, SDF-Diffusion Shim et al. (2023) exploited the two-stage framework using voxel-shaped SDF and proposed a method that generates high-quality 3D shapes by refining and upsampling coarse 3D shapes of low-resolution.

**3D losses**. Various loss functions have been proposed for training and evaluating 3D models. In particular, Chamfer Distance (CD) (Fan et al., 2017) that measures the average closest distance between pointclouds is reasonably good to regress vertices to their appropriate positions. However, it is usually insensitive to different density distributions. As a result, Tong et al. (2021) proposed Density-aware CD (DCD), a similarity metric derived from CD that can detect disparity of density distributions. But CD is not sufficient to produce high quality 3D meshes (Wang et al., 2018; Fan et al., 2017). Therefore, Wang et al. (2018) introduced a normal loss to characterize high order properties and favor smooth surfaces. However, Gkioxari Gkioxari et al. (2019) showed that minimizing chamfer and normal distances alone still results in degenerate meshes with optimization easily stuck in some local minimum. An edge loss (Gkioxari et al., 2019) and Laplacian loss Desbrun et al. (1999) are usually introduced to provide additional shape regularizers to encourage evenly sized mesh elements and smooth surfaces respectively. Current 3D models employ a multitude of loss functions. For instance, Pixel2Mesh (Wang et al., 2018) uses four types of pointcloud losses - *chamfer, normal, Laplacian and edge* - while MeshRCNN (Gkioxari et al., 2019) adopts a weighted sum of three pointcloud losses - *chamfer, normal and edge* and an $L_2$ loss over voxels. This requirement introduces challenges, complexity and design constraints into neural network architectures.

Beyond the aspects of storage and processing, meshes are typically best suited for downstream tasks. A rapidly growing application is 3D reconstruction of patient-specific geometries for medical simulations and analysis. 3D surface meshes are easily converted to meshes for finite element simulations, such as simulating the fractional flow reserve for coronary blockages, as pioneered by Heartflow (Nørgaard et al., 2022).

We also note that Physics Informed Neural Networks (PINNs) suffer catastrophically from not being invariant to translation, rotation, and scaling (Raissi et al., 2019). Although Group Equivariant neural networks (Cohen & Welling, 2016) could be used to tackle these issues, the restrictions to enforce make their application to PINNs challenging. The capabilities of the Zernike loss to be invariant to scaling, translation, and rotation make it particularly attractive for use in PINNs.

# B  ALGORITHMS

---

**Algorithm 1** Computation of $\mathcal{G}_{ijk}^{(S)}$

---

**Compute:** $\forall\, 0 \leq i,j,k \leq max_m,\ f_{ijk} = (i|j|k),$

$$s_{ijk} = \frac{1}{f_{ijk}(i+j+k+1)(i+j+k+2)}$$

**Define:**

$$conv(A,B) \quad = \quad einsum(A,K,K,K,B,\,'ijk,ail,bjm,ckn,lmn- \quad > \quad abc') \tag{24}$$

$\mathcal{G}_{ijk}^{(S)} \leftarrow 0$
**for** $face \in F$ **do** in parallel
    $\phi_a = V[F_{face}][a][:]\ \forall\, a\ \in \{1,2,3\}]$
    $P = (\phi_1 - \phi_3) \wedge (\phi_2 - \phi_3)$
    $\mathcal{C}_{ijk}^{(a)} = f_{ijk}x_a^i y_a^j z_a^k\ \forall\, i,j,k\ a \in \{1,2,3\}$
    $D = conv(C^{(2)}, C^{(3)})$
    $S = conv(C^{(1)}, D)$
    $\mathcal{G}_{ijk}^{(S)} +\!= S_{ijk}s_{ijk}\|P\|$
**end for**

---

---

**Algorithm 2** Computation of derivatives of $\mathcal{L}_{\mathcal{M}}$

---

**Given:** $\mathcal{G}_{i,j,k}$ and $\tilde{\mathcal{G}}_{ijk}$ (Ground truth geom. moments)
**Compute:** $\forall\, 0 \leq i, j, k \leq max_m,\ f_{ijk} = (i|j|k)$,
$$s_{ijk} = \frac{1}{f_{ijk}(i+j+k+1)(i+j+k+2)}, \tilde{\phi}_a^{(1)}, \tilde{\phi}_a^{(1)}$$
**Define:**

$$conv(A, B) = einsum(A, K, K, K, B,$$
$$'ijk, ail, bjm, ckn, lmn- > abc')$$

$\frac{\partial \mathcal{L}}{\partial V} \leftarrow 0$
**for** $face \in F$ **do** in parallel
    $\phi_a = V[F_{face}][a][:] \,\forall\, a \in \{1, 2, 3\}]$
    $\phi_{13} = (\phi_1 - \phi_3)$
    $\phi_{23} = (\phi_2 - \phi_3)$
    $P = (\phi_1 - \phi_3) \wedge (\phi_2 - \phi_3)$
    Compute $\frac{\partial P}{\partial \phi_i} \,\forall\, \phi_i$ (Eq. 22)
    $\frac{\partial \|P\|}{\partial \phi_a} = einsum(\frac{\partial P}{\partial \phi_a}, P, 'Pap, P- > ap')/\|P\|$
    $C_{ijk}^{(a)} = f_{ijk} x_a^i y_a^j z_a^k \,\forall\, i, j, k \quad a \in \{1, 2, 3\}$
    $D = conv(C^{(2)}, C^{(3)})$
    $S = conv(C^{(1)}, D)$
    Compute $\frac{\partial C_{ijk}^{(1)}}{\partial \phi_1}, \frac{\partial C_{ijk}^{(2)}}{\partial \phi_2}, \frac{\partial C_{ijk}^{(3)}}{\partial \phi_3}$ (Eq. 20)
    $D_1 = conv(C^{(2)}, C^{(3)})$
    $D_2 = conv(\frac{\partial C_{ijk}^{(2)}}{\partial \phi_2}, C^{(3)})$
    $D_3 = conv(C^{(2)}, \frac{\partial C_{ijk}^{(3)}}{\partial \phi_3})$
    $S_1 = conv(\frac{\partial C_{ijk}^{(1)}}{\partial \phi_1}, D_1)$
    $S_2 = conv(C^{(1)}, D_2)$
    $S_3 = conv(C^{(1)}, D_3)$
    $\frac{\partial S_{ijk}}{\partial \phi_a} = S_a \times s_{ijk} \,\forall\, a \in 1, 2, 3$
    $\frac{\partial \mathcal{G}_{ijk}}{\partial \phi_a} = \|P\|_2 \frac{\partial S_{ijk}}{\partial \phi_a} + \frac{\partial \|P\|}{\partial \phi_a} S_{ijk} \,\forall\, a \in 1, 2, 3$
    Compute $\frac{\partial \mathcal{L}^{face}}{\phi_a}$ (Eq. 17)
    Collect $\frac{\partial \mathcal{L}^{face}}{\phi_a}$ into $\frac{\partial \mathcal{L}^{face}}{\partial V}$
    $\frac{\partial \mathcal{L}}{\partial V} += \frac{\partial \mathcal{L}^{face}}{\partial V}$
**end for**

---

