# OpenReview forum: "MeshMoment: A Moment-Based Loss Function for 3D Learning on Meshes"
_ICLR.cc/2026/Conference — Submitted to ICLR 2026_

### Official Review · Reviewer_haw2 · 2025-10-22

**Soundness:** 1
**Presentation:** 2
**Contribution:** 2
**Rating:** 2
**Confidence:** 3

**Summary:**

This paper proposes a loss function for 3D learning on meshes that is based on comparing the geometric moments (GMs) of two meshes.
To make the loss function suitable for 3D learning, previous work the computation of GMs was optimized for computation on GPUs.
Furthermore, special care was taken to provide an efficient backward pass for gradient-based optimization.
The proposed loss function was evaluated in terms of its computational efficiency w.r.t. number of faces and moment order.

**Strengths:**

The strength of this paper in the proposal of a computationally efficient loss function that is robust to different triangulations and rigid transformations, unlike commonly used objective functions.

**Weaknesses:**

While the proposed loss function is evaluated in terms of computational efficiency, there is no evaluation of its performance in a real 3D learning or shape fitting tasks.
Given that the computed moments with truncated order are not necessarily unique to a shape, i.e. multiple shapes could have the same or similar moments, it is unclear how well the proposed loss function performs in practice.
A direct comparison with other commonly used loss functions in 3D learning tasks would be beneficial to support the claims of the paper.

On a more minor note, the typesetting in the equations could be improved for better readability.
E.g. one could wrap names of variables with \text{} or \mathrm{}.
Furthermore, a clearer reference to algorithms or figures in the text would help the reader (e.g. in lines 402 and 427).
Overall the paper presentation is not very polished (e.g. in line 101 a closing bracket is missing).

**Questions:**

* What is the minimum order of moments required to achieve reasonable "uniqueness" and good shape approximation/fitting performance in practice?
* How does the proposed loss function compare to commonly used loss functions in 3D learning tasks, e.g. Chamfer distance or Earth mover's distance in terms of shape fitting performance and computational efficiency?
* How sensitive is the proposed loss function to noise in the input meshes?

---

### Official Review · Reviewer_P55Z · 2025-10-28

**Soundness:** 2
**Presentation:** 1
**Contribution:** 2
**Rating:** 2
**Confidence:** 3

**Summary:**

This paper introduces a new metric, Geometric Moments (GMs), for measuring the similarity between 3D objects, with applications in evaluating reconstruction accuracy. Unlike the well-known Chamfer distance, which requires sampling point clouds from meshes before computation, the proposed GMs can be computed directly from mesh representations of surfaces, without relying on intermediate representations such as point clouds or voxel grids.

**Strengths:**

1. The paper proposes a novel metric for surface reconstruction and accuracy evaluation that operates directly on mesh representations, eliminating the need for point cloud sampling and neighborhood search as required by Chamfer distance.
2. The paper provides numerical experiments and analyses on the efficiency, scaling behavior, and gradient computation cost of the proposed Geometric Moments metric.

**Weaknesses:**

1. The paper is difficult to follow. The authors frequently discuss the orthogonal extension of Geometric Moments (GMs), namely Zernike Moments (ZMs), in the Introduction, Section 3 (Lines 196–210), and Section 5. However, this advanced variant is not actually explored due to the practical challenges mentioned in Lines 196–210.
2. Since the proposed metric targets surface reconstruction and its evaluation, it would be more convincing to demonstrate its benefits on standard reconstruction benchmarks and compare it against the widely adopted Chamfer distance. At present, the experiments remain limited to theoretical numerical analyses, which are too primitive to illustrate practical applicability.
3. The detailed calculus and derivations could be moved to the supplementary material to improve readability.
4. While Chamfer distance involves nearest neighbor search between two point clouds, its computational cost is not naively O(N^2). Modern data structure based techniques such as KD-trees significantly reduce runtime. The authors are encouraged to benchmark the construction time of their GMs against the sampling plus nearest neighbor search time required for Chamfer distance.

**Questions:**

How does the number of mesh vertices and faces affect the proposed metric?

---

### Official Review · Reviewer_5p8q · 2025-10-30

**Soundness:** 2
**Presentation:** 2
**Contribution:** 2
**Rating:** 2
**Confidence:** 3

**Summary:**

This paper proposes "MeshMoment," a loss function for 3D machine learning tasks that operates natively on mesh representations. The authors highlight the significant drawbacks of existing methods that rely on intermediate representations, such as the inconsistency and computational cost of point cloud-based losses (e.g., Chamfer distance). The core of the paper is an investigation into the efficient, vectorized computation of geometric moments (GMs) on GPUs, along with a manually derived, efficient backward pass (gradient) to make the loss differentiable and suitable for training. The paper provides a detailed performance analysis of this implementation, showing its scalability with respect to mesh face count and moment order.

**Strengths:**

Clear Problem Definition: The paper does an excellent job of identifying and clearly demonstrating a significant problem in 3D deep learning: the lack of a consistent, native loss function for mesh data. The critique of Chamfer distance, supported by Figure 1b, is sharp and well-articulated.

Thorough Performance Analysis: The technical investigation into the computational performance of different moment calculation algorithms (Pozo vs. Koehl) is thorough. The detailed scaling analysis (Figs 2, 3, 4) of the proposed vectorized forward and backward passes on modern GPUs is a solid piece of engineering work.

Valuable Direction: The motivation to find a geometry-aware, invariant-rich loss function for meshes is highly relevant, especially for applications like 3D reconstruction and scientific machine learning (e.g., PINNs), where current metrics fall short.

**Weaknesses:**

1. Lack of Novelty: The primary contribution appears to be an engineering one. The proposed loss function is based on the calculation of geometric moments, which itself is not new. The main "novelty" is the application of modern GPU vectorization and the implementation of a differentiable backward pass. While a useful engineering effort, this feels more like an implementation of a known technique rather than a fundamental research contribution for a top-tier conference.
2. Critical Lack of Application Experiments: This is the most significant flaw in the paper. The authors propose a loss function for optimization but provide zero experiments that actually use it for optimization. The paper presents a tool but never shows it being used for its intended purpose.
There are no 3D reconstruction, generation, or even simple fitting tasks.
A minimal "hello world" experiment would have been to take a source mesh and use the MeshMoment loss to optimize its vertices to match a target mesh. Without this, it's impossible to assess if the loss is actually useful in practice.
All claims about the loss's properties (e.g., consistency, geometry-awareness from Table 1) remain theoretical and unproven in an actual learning setup.
3. Unconvincing Application Prospects & Internal Contradictions: The paper fails to make a convincing case for its own contribution. The authors themselves explicitly state in the conclusion that geometric moments (GMs)—the very thing the entire paper is based on—"are not orthogonal, and therefore not suitable for optimization e.g. as a loss function."
This statement fundamentally undermines the paper's entire premise. The authors are, in effect, arguing against their own method.
They suggest that orthogonal moments (like Zernike moments, ZMs) would be better, but relegate this to "future work." This makes the current submission feel incomplete and premature. Why present a detailed analysis of a method you yourself admit is unsuitable?

**Questions:**

See Weaknesses section.

---

### Official Review · Reviewer_F6tJ · 2025-10-30

**Soundness:** 1
**Presentation:** 1
**Contribution:** 1
**Rating:** 2
**Confidence:** 2

**Summary:**

Paper introduces a new loss function for meshes that utilises
geometric moments. Authors provide a description of their vectorized implementation of moment computation and derivations of gradients
that allows for efficient computation w/o AD.
Evaluation of scaling of implementation is provided (no details
on bechmark).

**Strengths:**

*Significance:*
- This paper addresses an important problem of loss functions for meshes.
Moment loss seems to be indeed having multiple desirable properties
compared to commonly used chamfer distrances.

*Evaluation:*
- Provided implementation seems to be scaling polynomially according
to the figures (although those are non-trivial to interpret).

**Weaknesses:**

*Novelty:*
- The algorithms for moment computation exist, and the paper focuses exclusively on the implementation of those in a vectorized form.

*Evaluation:*
- There is no evaluation of the proposed loss function. Without it, it is unclear to me whether the implementation presented is in this paper brings any practical value.

*Writing:*
- The paper is at times hard to follow: for example, even the metric used in numerical evaluation is not properly introduced.
- Manuscript contains significant number of typos, and poor formatting (e.g. figure captions, missing entity references). The verbose derivations on P7 can be put into supplementary.

**Questions:**

1. Why there is no evaluation of the use of the loss and proposed implementation for an actual task / on a bechmark?

---

### Meta-Review · Area_Chair_uKdD · 2026-01-06

**Summary:**

The consensus among the four reviewers is rejecting the paper (all gave a rating of 2), primarily centered on the novelty of the paper paper, the lack of application experiments, and hard to follow due to the poor presentation.

The authors did not provide a rebuttal. Based on all of these information, I suggest rejecting the paper.

**Reviewer Concerns:**

None of the concerns have been addressed.

**Reviewer Scores:**

None of the reviewers will change their score since the authors did not provide a rebuttal.

---

### Decision · Program_Chairs · 2026-01-26

Reject